



# PM₁ composition and source apportionment at two sites in Delhi, India across multiple seasons

Ernesto Reyes-Villegas[1,*], Upasana Panda[2], Eoghan Darbyshire[1,Δ], James M. Cash[3,4], Rutambhara Joshi[1], Ben Langford[3], Chiara F. Di Marco[3], Neil Mullinger[3], W. Joe F. Acton[5], Will Drysdale[6], Eiko Nemitz[3], Michael Flynn[1], Aristeidis Voliotis[1], Gordon McFiggans[1,*], Hugh Coe[1], James Lee[6], C. Nicholas Hewitt[5], Mathew R. Heal[4], Sachin S Gunthe[2], Shivani[7], Ranu Gadi[7], Siddhartha Singh[8], Vijay Soni[8], James D. Allan[1,9,*]

[1]Department of Earth and Environmental Sciences, University of Manchester, Manchester, M13 9PL, UK
[2]Indian Institute of Technology Madras, Chennai, 600 036, India
[3]UK Centre for Ecology & Hydrology, Edinburgh Research Station, Penicuik, EH26 0QB, UK
[4]School of Chemistry, University of Edinburgh, Edinburgh, EH9 3FJ, Edinburgh, UK
[5]Lancaster Environment Centre, Lancaster University, Lancaster, LA1 4YQ, UK
[6]Wolfson Atmospheric Chemistry Laboratory, University of York, York, YO10 5DD, UK
[7]Department of Applied Sciences and Humanities, Indira Gandhi Delhi Technical University for Women, Delhi, 110006, India
[8]India Meteorology Department, New Delhi, 110003, India
[9]National Centre for Atmospheric Science, University of Manchester, Manchester, M13 9PL, UK

[Δ] Now at: The conflict and Environment Observatory, Hebden Bridge, HX7 5HZ, UK

*Correspondence to*: James Allan (James.Allan@manchester.ac.uk), Gordon McFiggans (g.mcfiggans@manchester.ac.uk) and Ernesto Reyes-Villegas (ernesto.reyesvillegas@manchester.ac.uk)

**Abstract.**

Air pollution in urban environments has been shown to have a negative impact on air quality and human health, particularly in megacities. Over recent decades, Delhi, India has suffered high atmospheric pollution, with significant particulate matter (PM) concentrations as result of anthropogenic activities. Organic aerosols (OA) are composed of thousands of different chemical species and are one of the main constituents of submicron particles. However, quantitative knowledge of OA composition, their sources and processes in urban environments is still limited. This is important particularly in India, as Delhi is a massive, inhomogeneous conurbation, which we would expect that the apportionment and concentrations will vary depending on where in Delhi the measurements/source apportionment is performed, indicating the need of multi-site measurements. This study presents the first multisite analysis carried out in India over different seasons, with a focus on identifying OA sources. The measurements were taken during 2018 at two sites in Delhi, India. One site was located at the India Meteorological Department, New Delhi (ND). The other site was located at the Indira Gandhi Delhi Technical University for Women, Old Delhi (OD). Non-refractory submicron aerosol (NR-PM₁) concentrations (ammonium, nitrate, sulphate, chloride and organic aerosols) of four aerosol mass spectrometers were analysed. Collocated measurements of VOC, black carbon, NOₓ and CO were performed. Positive matrix factorization (PMF) analysis was performed to separate the organic



fraction, identifying a number of conventional factors: hydrocarbon-like OA (HOA) related to traffic emissions, biomass burning OA (BBOA), cooking OA (COA) and secondary OA (SOA).

A composition-based estimate of $PM_1$ is defined by combining BC and NR-$PM_1$ (C-$PM_1$ = BC + NR-PM1). No significant difference was observed on C-$PM_1$ concentrations between sites; OD (142 +/- 117 $\mu g.m^{-3}$) compared to ND (123 +/- 71 $\mu g.m^{-3}$), from post-monsoon measurements. A wider variability was observed between seasons, where pre-monsoon and monsoon

showed C-PM1 concentrations lower than 60 $\mu g.m^{-3}$. A seasonal variation in C-$PM_1$ composition was observed; $SO_4^{2-}$ showed a high contribution over pre-monsoon and monsoon seasons while $NO_3^-$ and $Cl^-$ had a higher contribution in winter and post-monsoon. The main primary aerosol source was from traffic, which is consistent with the PMF analysis and aethalometer model analysis. Thus, in order to reduce $PM_1$ concentrations in Delhi through local emission controls traffic emissions control offers the greatest opportunity. PMF-AMS mass spectra will help to improve future aerosol source apportionment studies. The

information generated in this study increases our understanding of $PM_1$ composition and OA sources in Delhi, India. Furthermore, the scientific findings provide significant information to strengthen legislation that aims to improve air quality in India.

**Keywords:** AMS, PMF, on-line measurements, air quality, aerosol sources, Urban air pollution.







## 1 Introduction

Air pollution in urban environments has been shown to have a negative impact on human health, particularly in megacities. The World Health Organization (WHO) stated that in 2016 about 90% of the global population living in urban environments was exposed to particulate matter concentrations exceeding the WHO air quality guidelines (https://www.who.int/gho/phe/air_pollution_pm25_concentrations/en/, accessed 20/06/2019). According to the WHO global air quality database 2018, Delhi was ranked in 11[th] position of the cities with high particulate mass, measured as the mass of

particles whose size is equal to or lower than 2.5 µm ($PM_{2.5}$), having an annual average concentration of 143 µg.m$^{-3}$. Other Indian cities are in the top 20 with Varanasi being in first place with a $PM_{2.5}$ concentration of 217 µg.m$^{-3}$ (https://www.who.int/airpollution/data/cities/en/, accessed 17/06/2019). Previous studies have identified ambient fine particle concentrations to have detrimental effects on health, particularly the submicron fraction lower than or equal to 1 µm ($PM_1$) (Pope et al., 2002;Ramgolam et al., 2009).


Organic aerosol (OA) constitute a large proportion of $PM_1$ mass, with fractions between 40-60% in urban environments (Zhang et al., 2007). OA is composed of thousands of different chemical species and is difficult to study due to a variety of sources and processes. Black carbon (BC), another important component of $PM_1$, is a product of incomplete combustion and is recognised as one of the main climate-forcing components (Bond and Bergstrom, 2006;Lack et al., 2014). Moreover, the

WHO, in the health effects of BC report, states that while there was no clear evidence of BC directly affecting human health, it may however work as a medium to transport a wide variety of chemical species of varying toxicities (Janssen et al., 2012). BC sources and concentrations have been studied in India (Thamban et al., 2017;Jain et al., 2018a). In 2014, Gupta et al. (2017) studied the seasonal variation of BC in Agra, identifying biomass burning to be the major BC source, especially during winter. The BC mixing state was measured in a clean site (Nainital) and a polluted site (Gurgaon), with average BC concentrations of

1.0 and 11.0 µg.m$^{-3}$, respectively (Raatikainen et al., 2017). In 2016, a BC network in India was initiated with 16 Aethalometers (LASKAR et al., 2016); the availability of these instruments allowed the study of BC sources in different environments (Rajesh and Ramachandran, 2017;Kolhe et al., 2018;Nazeer Hussain et al., 2018).

Aerosol mass spectrometer (AMS) instruments (Aerodyne Research, Billerica, MA) have been widely used in different

locations around the world to quantify online real-time non-refractory $PM_1$ (NR-$PM_1$): OA, nitrate ($NO_3^-$), sulphate ($SO_4^{2-}$), ammonium ($NH_4^+$) and chloride ($Cl^-$). In India, the first NR-$PM_1$ measurements were taken on campus at the Indian Institute of Technology (IIT) in Kanpur in winter 2011, where OA was found to contribute 70% to NR-PM1 concentrations. (Chakraborty et al., 2015). Subsequently, a number of studies have been conducted at the same site showing dominance of the OA fraction with substantial contributions from secondary sources (Kumar et al., 2016) and looking at organo-nitrates and

OA sources (Chakraborty et al., 2016;Chakraborty et al., 2018) December 2015 to January 2016, where NR-$PM_1$ measurements helped to analyse the temporal characteristics of brown carbon (Satish et al., 2017). Aircraft measurements during the pre-





monsoon and monsoon periods in 2016, covering the NE Bay of Bengal and IGP regions, allowed the vertical and horizontal aerosol chemical composition to be studied (Brooks et al., 2019). Recent long-term measurements in Mahabaleshwar (Mukherjee et al., 2018), Bhubaneswar (Kompalli et al., 2020) and New Delhi (Gani et al., 2019) have allowed the study of NR-PM$_1$ seasonal variability.

Receptor modelling tools have been used in aerosol source apportionment studies in India, such as chemical mass balance, principal component analysis and positive matrix factorization (PMF) (Tiwari et al., 2013;Bhuyan et al., 2018;Jain et al., 2018b;Gadi et al., 2019;Shivani et al., 2019;Jain et al., 2020). PMF has been shown to be a useful tool to determine OA sources from aerosol mass spectrometer measurements across the northern hemisphere (Ng et al., 2010). In India, source apportionment studies using PMF have identified different OA sources, for instance, seasonal analysis in 2016-2017 in Mahabaleshwar, a high altitude site, identified hydrocarbon-like OA (HOA), biomass burning OA (BBOA) and two different types of OOA as the OA sources (Mukherjee et al., 2018;Singla et al., 2019). Similar OA factors were identified in a recent study performed in New Delhi, with the main OA sources being HOA, BBOA, and OOA from a 1 year analysis (Bhandari et al., 2019).

This work is part of the Atmospheric Pollution and Human Health in an Indian Megacity (APHH-India) programme (https://www.urbanair-india.org/). The main objective of the programme is to support research on the sources and emissions of urban air pollution in New Delhi, India, the processes underlying and impacting on these, and how air pollution then impacts on health. In agreement with two of the working packages of the programme (DelhiFlux and PROMOTE), the objectives in this work are:

- To study the aerosol concentrations in high-time resolution in Old and New Delhi through the year in two different locations.
- To identify main OA sources and to determine the contribution of primary and secondary OA.
- To study the interaction of aerosol concentrations with local meteorology
- To provide information to be used by the modelling community for improvements in the emission inventories, air pollution forecast and source apportionment.



## 2      Methodology

### 2.1      Site description and meteorology

The measurements were taken during 2018 at two sites in Delhi, India. One site was located at the Indian Meteorological Department (IMD), Mausam Bhawan, Lodhi Road, New Delhi. Lat 28.588, Lon 77.217, hereafter ND. The other site was located at the Indira Gandhi Delhi Technical University for Women. (IGDTUW) at New Church Rd, Kashmere Gate, Old Delhi. Lat 28.664, Lon 77.232, hereafter OD. Section S1 in the Supplementary Material shows the site locations.

The meteorology in India is highly influenced by the monsoon season, which takes places between June and September and is characterised by an increase in precipitation (Turner et al., 2019). Recent long-term measurements in Mahabaleshwar (Mukherjee et al., 2018) and New Delhi (Gani et al., 2019) have identified a clear pattern over the pre monsoon, monsoon and post monsoon seasons. Airborne measurements have studied the monsoon progression across the Indo-Gangetic Plain region and its impacts on aerosol characteristics (Brooks et al., 2019). In the present study, temperature showed a wide range of variability; winter was cool with an average temperature of 20 ℃, whereas pre-monsoon, monsoon and post-monsoon seasons were characterised by average temperatures of 36, 31 and 25 ℃, respectively. As expected, relative humidity (RH) showed the highest average over monsoon (76%) compared to the other seasons with RH lower than 60 %. Similar temperatures have been previously observed by Brooks et al. (2019) with 31 ℃ in pre-monsoon and 22° C in monsoon as well as by Gani et al. (2019) with temperatures between 10-20 ℃ in winter, 25-40 ℃ in pre-monsoon and 25-35 ℃ in monsoon. Figure S3 in supplement shows box plots of temperature, wind speed and RH for the different seasons.

### 2.2      Instrumentation

Four Aerodyne Aerosol Mass Spectrometers were deployed across the different seasons in 2018 (Table 1). One Aerosol Chemical Speciation Monitor (ACSM) (Ng et al., 2011), one compact time-of-flight Aerosol Mass Spectrometer (cToF-AMS) (Drewnick et al., 2005) and two High-Resolution Time-of-Flight AMS (HR-ToF-AMS) (DeCarlo et al., 2006). The principle of operation of aerosol mass spectrometer instruments has been widely explained in previous publications; in general these instruments quantify real time concentrations of NR-PM$_1$ (OA, NO$_3^-$, SO$_4^{2-}$, NH$_4^+$ and Cl$^-$) by vaporising the aerosols at 600 ℃ with electron impact ionisation at 70 eV and final detection in the mass spectrometer. The main difference between the instruments used in this study is the mass spectrometer, with differences in the sensitivity and the mass to charge (m/z) resolution. However, all are reliable instruments and successful intercomparisons with parallel measurements have been performed in previous studies (Ng et al., 2011;Crenn et al., 2015). In this study, the cToF-AMS and the HR-AMS performed parallel ambient measurements in OD during the pre-monsoon season, obtaining a successful intercomparison (Figure S5). Section S2 shows the calibrations and quality assurance analysis performed to the AMS measurements. The measurements in





PostM_OD_T and PostM_ND_T, were taken from a height of 32 m from a tower (T). The tower was used to perform eddy-covariance flux measurements, which will be reported elsewhere. The rest were ground-based measurements (~3 m high). The AMS measurements during the Diwali festival (7th November) for the period PostM_ND_T_C were lost due to a power cut, 170 thus this special episode was removed from the NR-PM$_1$ analysis.

**Table 1. Measurement dates and instrumentation. Winter (Win), pre-monsoon (PreM), monsoon (Mon) and post-monsoon (PostM). New Delhi (ND) and Old Delhi (OD) sites. ACSM (A), two HR-ToF-AMS instruments were used (H1 and H2), cToF-AMS (C). Eddy-covariance flux measurements tower (T).**

| Instrument | Event | start | end | Days | Feb | Mar | Apr | May | Jun | Jul | Aug | Sep | Oct | Nov | Dec |
|---|---|---|---|---|---|---|---|---|---|---|---|---|---|---|---|
| ACSM | Win_ND_A | 05/02 | 03/03 | 26 | ▓ | | | | | | | | | | |
| HR-AMS_1 | PreM_ND_H1 | 24/04 | 30/05 | 36 | | | ▓ | | | | | | | | |
| cToF-AMS | PreM_OD_C | 28/05 | 09/06 | 12 | | | | ▓ | | | | | | | |
| HR-AMS_2 | PreM_OD_H2 | 26/05 | 28/06 | 33 | | | | ▓ | | | | | | | |
| HR-AMS_2 | Mon_OD_H2 | 03/08 | 19/08 | 16 | | | | | | | ▓ | | | | |
| HR-AMS_2 | PostM_OD_H2 | 09/10 | 04/11 | 26 | | | | | | | | | ▓ | | |
| cToF-AMS | PostM_OD_C | 11/10 | 15/10 | 4 | | | | | | | | | ▓ | | |
| HR-AMS_2 | PostM_OD_T_H2 | 06/11 | 20/11 | 18 | | | | | | | | | | ▓ | |
| cToF-AMS | PostM_ND_T_C | 06/11 | 20/11 | 32 | | | | | | | | | | ▓ | |

175

Other instruments used along with the mass spectrometers include: two Aethalometers (Magee Scientific; one model AE31 and one model AE33) and one Multi-Angle Absorption Photometer (MAAP, Thermo Fisher Scientific) (Petzold et al., 2002). Table 1 shows details about the instrument locations and sampling periods. Meteorology data, apart from Planetary Boundary 180 Layer Height (PBLH), was downloaded from https://ncdc.noaa.gov/ (last access: 05/01/2019) at 1-hourly resolution. PBLH data sourced from ECMWF ERA5 0.25 deg. results in 1-hour resolution.

PMF (Paatero et al., 2002) was used to identify potential sources of OA, based on the Source Finder tool, SoFi 4.8 (Canonaco et al., 2013). Section S3 in supplement provides detailed information about the criteria to select the OA factors. The PMF analysis of the OD_H2 measurements presented in this work was performed, by Cash et al. (2020) for the complete dataset 185 (26/05 – 20/11) using high-resolution organic mass spectra. The PMF analysis to the PreM_ND_H1 dataset was performed to unit mass resolution data. BC concentrations measured with the Aethalometer AE-31 were corrected following the Weingartner method (Weingartner et al., 2003) and using the SP-2 as a reference BC measurement. The aethalometer model was applied, following the Sandradewi method (Sandradewi et al., 2008), to aethalometer measurements, both AE-31 and AE-33, to identify the contribution of biomass burning and fossil fuel to BC concentrations. Section S4 in the Supplement shows the BC data 190 processing.



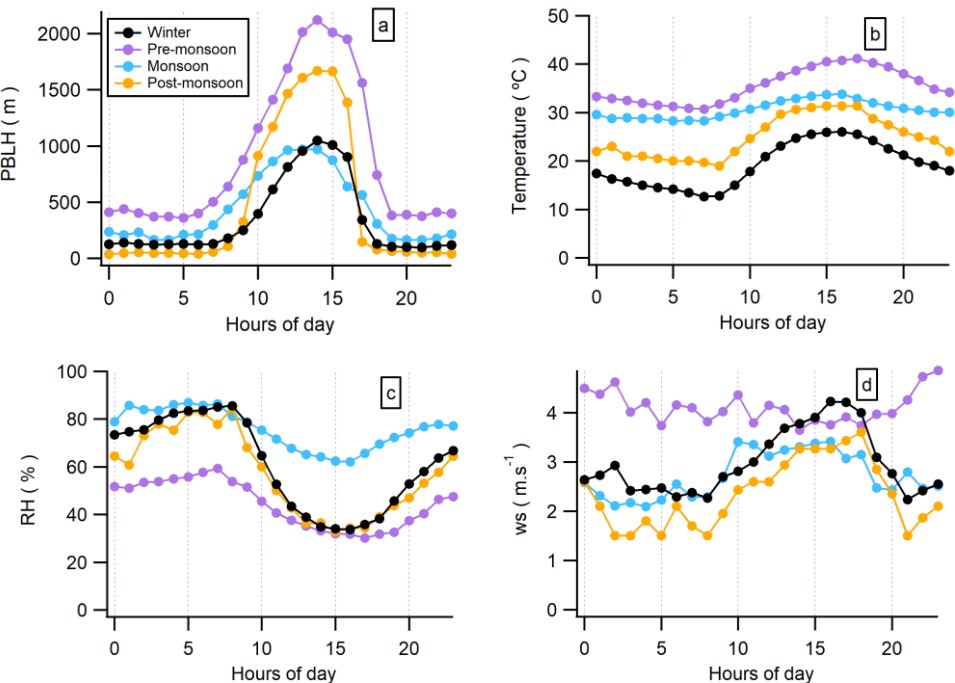

**Figure 1. Box plots with (a) planetary boundary layer height (PBLH), (b) temperature, (c) relative humidity and (d) wind speed for the different seasons. The marker represents the mean.**

## 3. Results

### 3.1 Aerosol concentrations and composition

Aerosol measurements were taken in OD and ND over different seasons in 2018. The highest concentrations were observed in the post-monsoon season, where the PostM_OD_H2 period reached OA of 400 µg.m$^{-3}$ and the following period, PostM_OD_T_H2, with peaked OA concentrations 630 and 1200 µg.m$^{-3}$. Low aerosol concentrations were observed in the monsoon season, where OA reached between 80 – 200 µg.m$^{-3}$. Figure S6 shows the time series of the various measurements. Similar to Gani et al. (2019), we combine BC and NR-PM$_1$ concentrations to develop a composition based estimate of PM$_1$ (C-PM$_1$ = BC + NR-PM$_1$). The season with the highest averaged C-PM$_1$ concentrations were observed in PostM-OD, with average concentrations (+/- SD) of 142 +/- 117 µg.m$^{-3}$ in PostM_OD_T_H2 (Fig. 2.a). Low aerosol concentrations were observed in pre-monsoon and seasons with C-PM$_1$ average concentrations lower that 40 µg.m$^{-3}$ in PreM_ND_H1, and Mon_OD_H2, with OA being the component with the leading contributor to PM$_1$.

Measurements during post-monsoon were taken in OD (PostM_OD_T_H2) and ND (PostM_ND_T_C) over the same time (6-20 Nov). This comparison shows average C-PM$_1$ concentrations in OD were slightly higher (142 +/- 117 µg.m$^{-3}$) compared to ND (123 +/- 71 µg.m$^{-3}$). It is worth mentioning that, while removing the Diwali Festival as a special event, the measurements





shown in this work still include high concentrations from festivities that took place before and after the Diwali Festival on November 7[th]. The ND site is located in affluent residential green area, where the transit of heavy good vehicles is controlled; thus, it could be expected that the use of fireworks is also restricted here, whereas in OD, there is less control of the firework
activities. While no significant differences were observed when comparing between sites, a clear difference between seasons was observed; pre-monsoon and monsoon with similar C-PM$_1$ concentrations lower than 60 µg.m$^{-3}$, while post-monsoon with C-PM$_1$ concentrations higher than 120 µg.m$^{-3}$. Detailed statistical analysis for NR-PM$_1$ and BC is presented in Table S3.

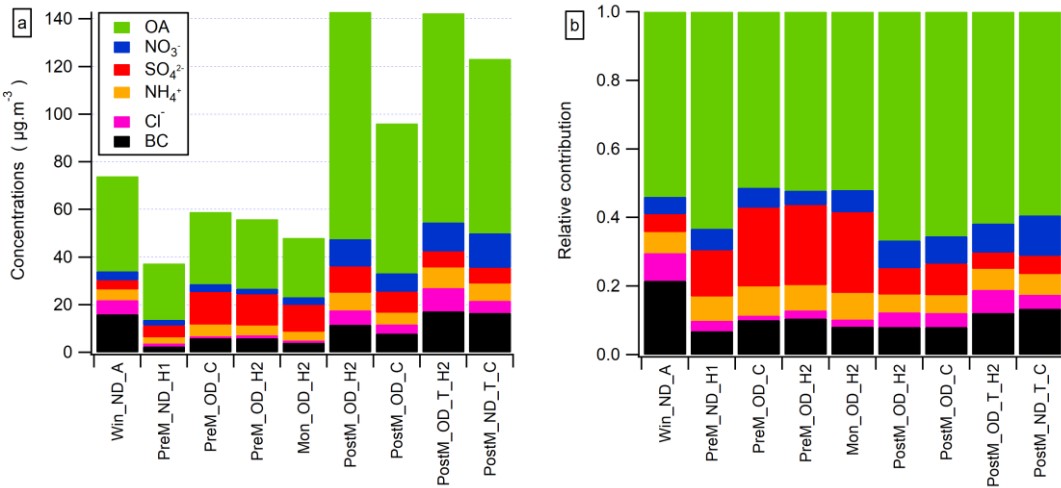

**Figure 2. Average aerosol concentrations (a) and relative contribution to C-PM1 (b).**

OA makes the main relative contribution to PM$_1$ with values of 0.45-0.62 followed by BC with values of 0.13-0.24 (Fig. 2.b). This high relative contribution of OA has been observed in previous studies in India (Bhandari et al., 2019;Singla et al., 2019). The high BC contributions of around 18%, suggest an important contribution from primary sources. Sulphate made a higher contribution in pre-monsoon and monsoon while NO$_3^-$ and Cl had a higher contribution in winter and post-monsoon, similar to a previous study performed in Delhi (Gani et al., 2019).

This wide range of concentrations is related to the meteorological conditions (Fig. 1), mainly with boundary layer height as a result of changes in temperature; winter and PostM-T had a temperature of 20-25º C, resulting in wind speeds of 2.5-3.0 m.s$^{-1}$, which may have resulted in a more stable mixing layer, accumulating aerosol concentrations. In early pre-monsoon, an average temperature of 35º C with wind speeds of 3.5 m.s$^{-1}$ facilitated greater aerosol dispersion. From the diurnal cycles (Fig.
3), it can be seen that NO$_3^-$ and Cl$^-$ showed high concentrations during the morning while BC and OA showed high concentrations at night. SO$_4^{2-}$ showed less diurnal variability, suggesting a regional origin with large sources at midday when the photochemistry takes place compensating for the boundary layer effect. BC is an aerosol constituent related to primary emissions, fossil fuel (mainly traffic) and biomass burning (Sandradewi et al., 2008), hence the fact that OA had a diurnal pattern similar to BC during the post-monsoon season suggests OA was potentially dominated by primary sources; while in





the pre-monsoon season, OA showed an increase in concentrations during the day, suggesting a larger contribution from
secondary OA (Shrivastava et al., 2017). This primary-secondary origin of OA will be analysed in the next section.

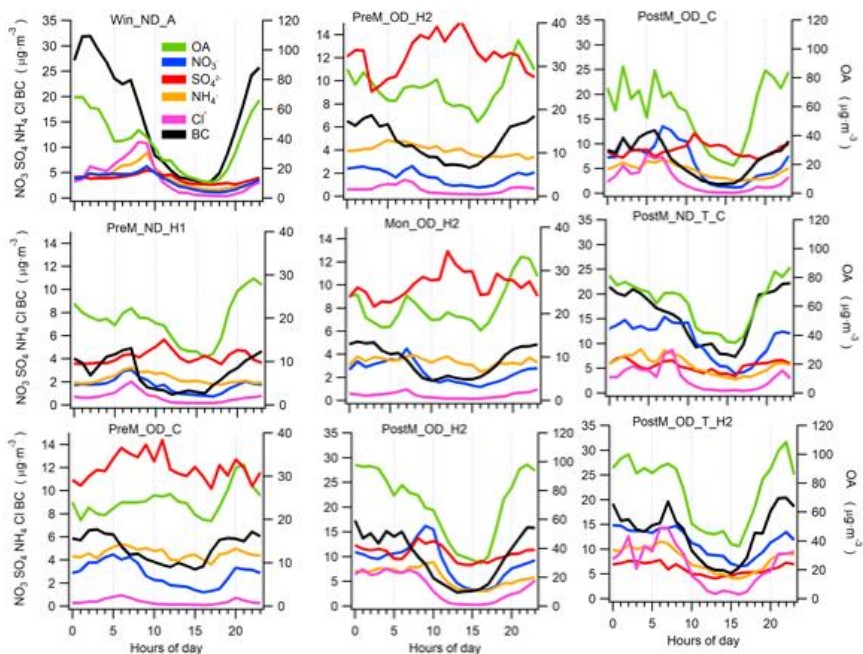

**Figure 3. Median diurnal concentrations of the aerosol chemical components**

## 240   3.2     Source apportionment of the organic aerosol

Organic aerosol is composed of thousands of chemical species, making source attribution challenging. Methods using receptor
modelling tools such as PMF are often employed to deconvolve OA into various OA factors that, depending on their
composition and temporal variations, can be attributed to potential OA sources. In this work a number of factors were identified
with PMF analysis (Fig. 4); three common factors were found in all the seasons, hydrocarbon-like OA (HOA), biomass burning

OA (BBOA) and low volatility oxygenated OA (LVOOA). Other identified sources were cooking OA (COA), semi-volatile
oxygenated OA (SVOOA) and oxygenated primary OA (oPOA). HOA is related to fossil fuel emissions, mainly attributed to
traffic, whose mass spectrum is characterised by peaks at mass to charge ratio (m/z) 55 and 57, and is dominated by the $C_xH_y^+$
family (Zhang et al., 2005). BBOA is characterised by m/z 60 ($C_2H_4O^+$) which is related to anhydrosugar fragments, such as
levoglucosan (Alfarra et al., 2007). The mass spectrum of COA presents peaks at m/z 55 and m/z 57, similar to the HOA

spectrum but with a lower peak at m/z 57 (Allan et al., 2010;Mohr et al., 2012). The two oxygenated OA (OOA) factors
identified LVOOA and SVOOA. LVOOA, with a dominant peak at m/z 44 ($CO_2^+$), and SVOOA, typically with a peak at m/z
44 and a larger peak at m/z 43 (mostly $C_2H_3^+$) compared to LVOOA (Zhang et al., 2011).





The HOA mass spectrum was consistent across the collection periods and between the different instruments deployed, with
peaks at m/z 41, 43, 55 and 57, and its diurnal cycle showed high concentrations in the morning and at night resulting from
the traffic rush hour. Moreover, HOA showed its highest concentrations during periods of low wind speeds (Fig. S16) and
shallow boundary-layers, highlighting the importance of HOA as a primary local source: even a small local source emitting
into a shallow nocturnal boundary-layer can result in significant accumulation of concentrations. The LVOOA mass spectrum
is also consistent over all the seasons with peaks at m/z 18 and m/z 44, its diurnal cycle shows the highest concentrations to
have been around 10:00 to 15:00 hrs, suggesting to be related to the stronger radiation occurring at that time, resulting in
secondary aerosol production from photochemistry. We can observe larger variability in the BBOA mass spectra, with changes
to the peak at m/z 60, which has a higher relative intensity over the post-monsoon and winter seasons. oPOA, which shows
peaks at m/z at 43 and m/z 44, has a characteristic diurnal pattern with high concentrations around 09:00 hrs, while the mass
spectrum resembles that of OOA, the diurnal cycle behaved different to those of SVOOA and LVOOA. A more detailed
analysis about its possible origin will be provided in the next section. COA was identified over per-monsoon and post-
monsoon, but not in PostM_ND_T or during winter. Moreover, previous studies in urban environments have identified a lunch
peak in the diurnal plots (Allan et al., 2010;Young et al., 2015). However, this peak was not clearly observed in any of the
seasons where COA was identified. Perhaps, the high OA concentrations resulting from the other anthropogenic emissions,
i.e. traffic and biomass burning along with the boundary layer effect, would favour the mixing of OA sources and PMF will
struggle to completely separate the OA sources (Lanz et al., 2008).

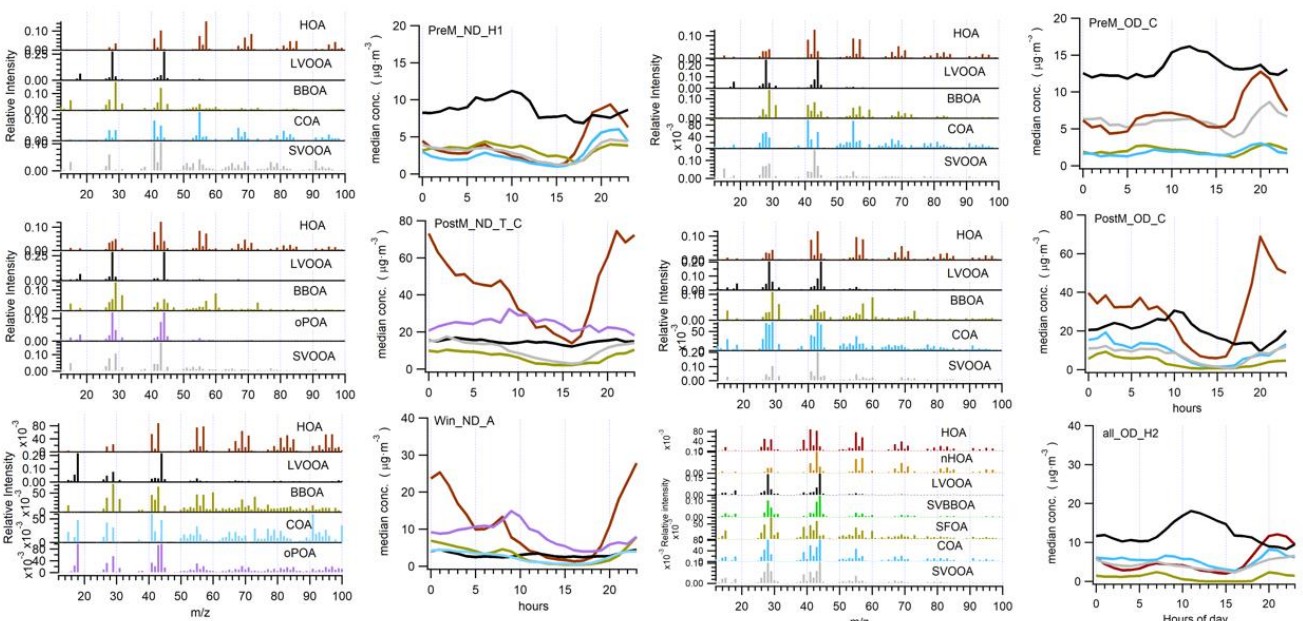

**Figure 4. Mass spectra and diurnal plots of OA factors from PMF analysis to the different periods. Here the period all_OD_H2 is
display to compare with the other periods. The period all_OD_H2, which shows the high resolution PMF analysis performed to the**





**complete dataset of AMS (H2) 26th of May – 23rd November 2018, is analysed in detail by Cash et al. (2020). Detail analysis of the all_OD_H2 dataset is presented in Table S4.**

## 4 Discussion

### 4.1 The role of meteorology on aerosol concentrations.

The meteorology in India has been summarised in Section 2.1, stating the impact of the monsoon season and the difference

between pre-monsoon and post-monsoon. This work allowed the study of NR-PM$_1$ seasonal variability, which showed strong variations through the seasons (Figure 2), where a seasonal cycle was observed with high aerosol concentrations in post-monsoon and winter with low temperatures. From the previous diurnal plots in winter_ND (Fig. 3), high Cl$^-$ concentrations were observed in the morning. PMF analysis performed on this dataset also identified a factor referred to as oPOA with a morning peak (Fig. 4.a). Figure 5 shows the diurnal trends of Cl and oPOA along with meteorology parameters. Both Cl and

oPOA peak around 09:00 hrs, at the time when RH shows its highest value and their concentrations start decreasing when the PBLH starts increasing around 10:00 hrs. This decrease in Cl and oPOA concentrations is perhaps due to mixing with fresh air masses from the break-up of the boundary layer, moreover, it is likely that the Cl- reduction is due to some repartitioning into the gas-phase as temperature increases. NH4Cl has a higher vapour pressure than NH4NO3 and will dissociate first. This is also consistent with the apparent deposition rate of Cl- increasing before that of NO3- increases in the morning. The same

high Cl concentrations were observed by Chakraborty et al. (2018) in NR-PM$_1$ measurements taken during Sep-Oct 2014 in Kanpur, India where similar meteorological conditions were present. This suggests the impact RH, PBLH and temperature have on Cl and oPOA concentrations. Figure S17 shows Cl and oPOA have different sources, it can be observed that Cl follows the trend of NH$_4^+$, which is expected as the AMS is sensitive mainly to NH$_4$Cl but not to other Cl$^-$ compounds, and oPOA follows the same pattern as SO$_4^{2-}$. It is possible that the properties of the condensed material in oPOA may be similar to the

assumed NH$_4$Cl, even though the sources are different, i.e. both have a similar volatility.

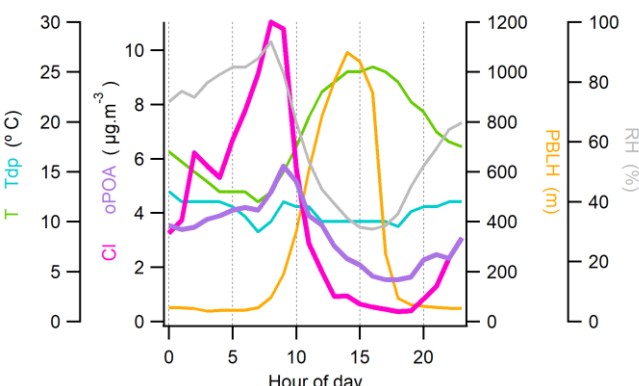

**Figure 5. Diurnal median values of meteorological parameters, Cl$^-$ and oPOA concentrations in Winter_ND**



## 4.2      Organic aerosol sources and their contribution to NR-PM₁.

Seasonal dependencies of the PMF factors can be investigated by looking at their average contributions (Fig. 6). HOA is the factor with the highest concentration with 6.0 – 55.0 µg.m$^{-3}$ for Pre-monsoon and Post-monsoon, respectively. Comparing these concentrations with the PostM_OD_H2 dataset (Table S4), HOA concentrations of 30 ug.m-3 were identified. These HOA concentrations represent a relative contribution to OA sources of 20 to 50%. Traditionally, HOA has been related mainly to traffic emissions of vehicles using petrol or diesel (Zhang et al., 2005;Platt et al., 2017). However, in Delhi, compressed

natural gas vehicles and generators might also contribute to HOA concentrations (Prakash et al., 2020). It is interesting to see that BBOA has low concentrations (2.0 – 8.0 µg.m$^{-3}$) in the ND and pre-monsoon seasons with a low contribution to OA sources (0.10 – 0.15). However, Cash et al. (2020), in the PostM_OD_T_H2 analysis, identified a BBOA average concentration of 48.22 µg.m$^{-3}$, representing a relative contribution of 25%. COA is an important primary source in OD, with average concentrations of 2.0 - 12.0 µg.m$^{-3}$. It is worth mentioning that COA concentrations might be overestimated due to a potential

higher relative ionisation efficiency, which has been observed previously in a laboratory experiment (Reyes-Villegas et al., 2018). Considering HOA, BBOA and COA as primary OA (POA) and the rest of the sources as OOA and looking at the POA relative contribution to OA concentrations (Fig. 6.d), there is a more constant concentration in ND with values between 54 – 56%, while POA relative contributions in OD range between 46 – 66%. These POA relative contributions close to 50% over different times of the year have been previously observed in a study performed in 2017 at IIT Delhi, New Delhi (Bhandari et

al., 2019), with similar average concentrations for POA and OOA (52 and 56 µg.m$^{-3}$ for winter as well as 30 and 31 µg.m$^{-3}$ for pre-monsoon, respectively). Cash et al. (2020) identified POA relative contributions of 45% and 52% for pre-monsoon and monsoon and 64% for post-monsoon, showing the importance of POA concentrations in OD during this season. OOA concentrations are produced by both POA and precursor gas. In other urban environments, it is possible to see a more variable POA-OOA ratio contribution to OA concentrations (Jimenez et al., 2009). For instance, in London, UK, in 2012 POA

contributions ranged from 80% in winter to 45% in summer (Young et al., 2015), perhaps as a result of the lower temperatures





during winter compared to summer along with less photochemistry in winter at higher altitudes. These findings demonstrate the impact primary $PM_1$ emissions in Delhi have on OA concentrations over the year.

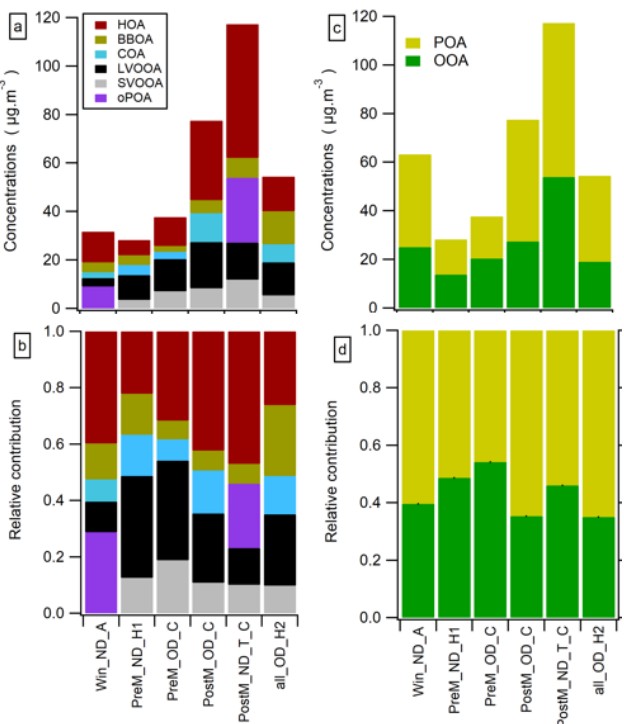

**Figure 6. OA average concentrations (6.a) and relative contribution (6.b). The whiskers represent the standard deviation of POA/OOA contribution after running PMF with different Fpeaks between -1 to 1 with steps of 0.2 to prove the stability of the separation.**

### 4.3    fossil fuel and biomass burning sources of BC.

The high contribution of POA has been explained in Section 4.2 and more specifically, traffic was identified as an important OA source. This finding is supported by the aethalometer model analysis. This model based on the Sandradewi method (Sandradewi et al., 2008) uses the absorption coefficient ($b_{abs}$) and the absorption angstrom exponent to apportion the contribution of fossil fuel ($b_{abs\_950ff}$) and biomass burning ($b_{abs\_470bb}$) to Aethalometer measurements (refer to Section S4 in the Supplement for details). Figure 7 shows the mean absorption coefficients for fossil fuel and biomass burning, where high $b_{abs\_950ff}$ values compared to $b_{abs\_470bb}$ are observed. Pre-monsoon and monsoon are the seasons with low biomass burning influence, with average values between 8 – 20 Mm$^{-1}$; during these seasons, fossil fuel showed $b_{abs}$ values between 35 and 50 Mm$^{-1}$. In post-monsoon the influence of biomass burning starts to increase with average values around 50 Mm$^{-1}$. However, at this time, traffic showed higher average values between 100 and 150 Mm$^{-1}$. In winter, biomass burning has an important influence, with an average of 80 Mm$^{-1}$, while fossil fuel still has a major influence with a value of 95 Mm$^{-1}$. It is only during





the Diwali festival, from OD-PostMon (Fig. S13, where biomass burning has a significant contribution to BC (700 Mm$^{-1}$).

However, during this period, there is an important contribution of BC from fossil fuel (500 Mm$^{-1}$), showing the importance of

fossil fuel as a BC source.

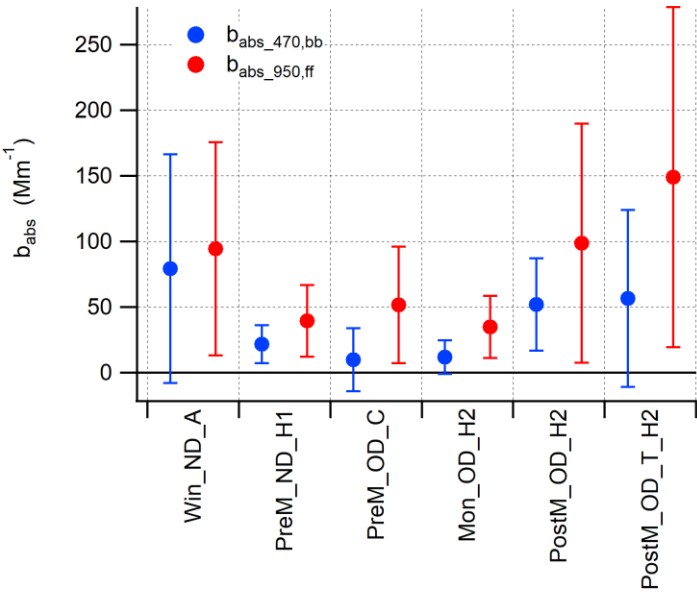

**Figure 7. Average absorption coefficients for fossil fuel (b$_{abs\_950ff}$) and biomass burning (b$_{abs\_470bb}$). The whiskers represent ± 1**
**standard deviation.**

## 5. Conclusions

Aerosol composition was studied to compare Old Delhi (OD) and New Delhi (ND) over different seasons in 2018 with a focus

on source apportionment. No significant difference was observed on C-PM$_1$ concentrations (defined as total AMS mass plus

black carbon, BC) from post-monsoon measurements between sites; OD (142 +/- 117 µg.m$^{-3}$) compared to ND (123 +/- 71

µg.m$^{-3}$). A wider variability was observed between seasons, where pre-monsoon and monsoon showed average C-PM$_1$

concentrations lower than 60 µg.m$^{-3}$. OD presented high SO$_4^{2-}$ concentrations (7.0 µg.m$^{-3}$ in OD and 3.4 µg.m$^{-3}$ in ND). In the

post-monsoon and winter seasons, the C-PM$_1$ mass was dominated by organic matter (OA) rather than inorganic and the largest

single contributor was hydrocarbon-like organic aerosol (HOA). NH$_4^+$, NO$_3^-$ and Cl$^-$ peaked in the morning while OA peaked

at night. A seasonal variation of PM$_1$ composition was observed; SO$_4^{2-}$ showed a high contribution over pre-monsoon and

monsoon seasons while (more volatile) NO$_3^-$ and Cl$^-$ made a larger contribution in the cooler seasons, winter and post-

monsoon. The fact that no remarkable difference between sites was found suggests the air pollution in Delhi is a regional



problem that is influenced by meteorological conditions over the year, being the post-monsoon and winter the seasons with highest pollutant concentrations.

Analysis of the organic mass spectra by positive matrix factorisation (PMF) identified a number of conventional factors: HOA, biomass burning OA (BBOA), cooking-like OA (COA), semi volatile oxygenated OA (SV-OOA) and low volatility oxygenated OA (LVOOA). One additional factor (oPOA) had a particular diurnal trend with its mass spectrum looking like OOA and behaving like $SO_4^{2-}$, suggesting being semi-volatile. However, it was not possible to determine its source. Traffic showed to be the main primary aerosol source for both OA and BC, as seen with the PMF analysis (HOA) and aethalometer

model analysis ($b_{abs\_950ff}$). The measurements indicate that in Delhi further control of primary traffic exhaust emissions would make a significant contribution to reducing $PM_1$ concentrations. Moreover, by controlling gas precursor emissions, OOA concentrations may also be reduced.

Primary organic aerosol (POA) made a relatively constant contribution at the different sites and seasons with a relative

contribution around 50 % to OA. This shows both the importance of primary sources and the fact that daily average temperatures remain above 20 ℃ throughout the year; 20 ℃ in winter and 43 ℃ in pre-monsoon, compared, for example, with European urban environments were average temperatures can drop to 5 ℃ or lower in winter and summer temperatures are around 25-30 ℃. This temperature might increase the regional pollutant transport allowing an even POA – SOA distribution over the Delhi region. When comparing these results with Cash et al. (2020), it shows the importance POA has in OD during

the post-monsoon season, where a POA relative contribution of 63% was observed. Meteorology played an important role in aerosol concentrations. High concentrations of $Cl^-$ and oPOA were observed in winter_ND which was characterised by high RH, with a sudden drop in concentrations when the boundary layer broke up in the morning and temperatures rose. This $Cl^-$ peak has been observed in a previous post-monsoon study in Kanpur India (Chakraborty et al., 2018;Gadi et al., 2019). More detailed research should be done in order to identify the potential sources/processes that caused the peak in OA concentrations

around 9-10 am in urban Indian environments, which has been observed taking place during post-monsoon and winter.

The information generated in this study increases our understanding of $PM_1$ composition and OA sources in Delhi, India. The PMF-AMS mass spectra can be used in future source apportionment studies in India and perhaps other urban environments to improve the identification of sources and provides a unique dataset for the assessment of atmospheric chemistry and transport

models. Furthermore, the scientific findings provide significant information to target legislation that aim to improve air quality in India.





**Acknowledgements**

This work was supported by the UK NERC through the DelhiFlux and PROMOTE projects under the Newton-Bhabha Fund Programme "Air Pollution and Human Health in a Developing Megacity (APHH-India)". NERC reference numbers: NE/P016502/1, NE/P01643X/1, NE/P016472/1 and NE/P016480/1. The monsoon measurements were supported by the NERC National Capability award SUNRISE (NE/R000131/1). James Cash is recipient of a NERC E[3] DTP studentship (NE/L002558/1).

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
