# Peer review of "PM1 composition and source apportionment at two sites in Delhi, India across multiple seasons"

_Atmospheric Chemistry and Physics, 2020_

## Referee Comment (RC1) · Anonymous Referee #1 · 14 Oct 2020

This is what I would consider a "standard" AMS and aethalometer study to document air pollution levels in a major urban area in a part of the world where air quality is among the worst; thus, it does fall within the scope of ACP. Its novelty lies in the application of proven methodology in a new location with a sparse amount of spatial and temporal resolution. It is not groundbreaking, but it is important, as the conclusions are relevant for air quality control efforts in what many consider the developing world. The question then becomes, however – could more be done with the data?

On the whole, it is well written and easy to follow (though some care needs to be paid to figure and table numbering), the abstract and title are accurate and the authors

use appropriate citations. The figures are easily read and understood – especially for members of the community who do this sort of work (diurnal profiles, time series, AMS PMF factor spectra).

My main concern and why I am unable to recommend publication at this time is the lack of evidence that the instruments are actually reporting comparable data – this is based on a lack of comparison of the AMS spectra and on the lack of agreement (approaching 25% difference) in the one set of collocated data shown. See below.

Specific comments

Line 154, should Figure 1 be called out here? In fact, Figure 1 is not called out at all until Line 226, though figures should be called out sequentially.

Line 165, the C and HR-MS comparison concentrations in S5 do not match the numbers given in the preceding text. In addition, they disagree by $\sim$6 ug/m3 (26 for C, 32 for HR), which is a 23% difference. Does this limit the robustness of the spatial comparison made? I actually think this is the most significant weakness of this manuscript. How can the authors justify comparing data from separate locations/times when two of the instruments do not agree when collocated? Were any comparisons done on the PMF results? Are their spectra similar? Do the PMF results actually indicate that the factors being compared have similar characteristics such as O:C, H:C, etc.? Before I can recommend this paper for publication, I need to be convinced that the data are actually comparable, not just told that they are, even if it is in the supplement (line 254). Simply showing the spectra in Figure 4 is insufficient in my opinion.

Line 179, should this be table S1, not table 1?

Line 201 (and 205 and other places), I would argue that the concentrations observed are lower, not low. 80-200 ug/m3 are still very large concentrations!

Line 245. I believe most AMS data are presented as LO-OOA and MO-OOA (less and more oxidized) rather than as SV and LV, respectively, in more recent literature.

Section 4.1. I find the discussion on oPOA to be fairly weak, with no justification of the suppositions made. It makes sense that meteorology impacts the oPOA dynamics, then it is mentioned that it tracks sulfate, then it is stated that oPOA may have a similar volatility as NH4Cl. No conclusions are made. I suggest this portion of the discussion be removed or strengthened considerably.

Supplement S2. Please clarify how CE was determined. The paper states that the authors used 0.5 based on the ACSM manual. However, in the plots, some show CE = 1, and others show CE = 0.5 (while others show CDCE). In addition, the x-axis in these plots show PM2.5, not PM1. Is that reasonable to use?

Supplement S3 and S4. Please check the figure numbering.

Supplement S5. Please correct the caption for S17, as it shows more than Cl and oPOA.

---

## Referee Comment (RC2) · Anonymous Referee #2 · 9 Jan 2021

The paper by Reyes-Villegas reports on measurements of aerosol composition and absorption at 2 sites in India (new Delhi and Old Delhi) where ground- based measurements were carried out in different seasons in 2018. Diurnal and seasonal trends in composition and organic aerosol types, resolved by positive matrix factorization, are reported. PM1 composition and concentration differences between the two sites in all seasons are minimal. Organic aerosols (OA) were the dominant component of PM1 in all seasons; higher concentrations of all species were measured in post-monsoon/winter, owing to changes in meteorological conditions. During pre-monsoon, oxygenated OA (LVOOA) was the dominant OA type with peaks appearing during the day, while in post-monsoon and winter periods, hydrocarbon-like OA (HOA) was the dominant aerosol type, with minimum concentrations during the day. Biomass burning OA, oxygenated primary OA, cooking OA, other oxygenated OA (SVOOA) were also identified although not consistently at all sites and all seasons (except for BBOA and COA). Given this is not the first set of measurements in the area (see lines 94-105 for a list of comparable studies) and lack of new scientific insights presented here, I suggest the manuscript be considered as a "Measurement Report". As such, I still have concerns about the analysis and data quality; some parts of the report also benefit from new reorganization and further clarifications.

Technical concerns:

-    What was the sampling protocol/set up at different sites? Was there any cyclone used on the inlet? Was there any control on humidity? If not, how could changes in ambient RH diurnally and between seasons impact the measurements?

-       Quantification of different AMSs and Figure S4: what is the difference between pre-flux and flux period? Why is the behavior of the comparisons so different between these periods when using the same CE correction factor (0.5 or CDCE)? For pre-flux period, slope with CE=0.5 and CDCE is very similar. It may be that the composition really dictates CE to be close to 0.5, but better justification/explanation needs to be provided as to why 0.5 is used for the whole study and not CDCE. Comparison is done on online PM1 and PM2.5 filter-based data. How/where are these filters collected? No information is provided, yet the measurements serve as the basis for justifying the selected value of the CE! What is the suspected contribution of 1-2.5 um particles to mass that justifies the comparison?

-       Aethalometer data and contribution of BB vs. traffic: it was not explained why $AAE_{wB}=2$ and $AAE_{tr}=0.8$ is selected. It seems the basis for selecting these factors is the match between the traces in Fig. S13 for the time period in the box, but it wasn't explained why is it expected to have equal absorption from wood burning and traffic at two different wavelengths during Diwali period to justify selecting the default AAEs? I also disagree that there was no change in the absorption calculations using different values of AAE; clearly the b_abs_470wb changes in other panels of this graph depending on the AAE value. Why was a different $AAE_{tr}$ used for AE-33 ND-Winter period (Fig. S15)? In general, I'm also not sure the way aethalometer data are presented is useful.  What would be more useful could be the fraction of absorption at 470 nm that is due to wb vs. traffic and not the absolute value of absorption due to wb at 470 nm.

-       References to PMF factors for oxygenated OA are given in terms of expected volatility (LV/SVOOA). Isn't it more accurate to call them based on their oxygenation level since the AMS doesn't measure volatility directly? Factors show up in the last panel of Fig. 4 (SFOA and nHOA) that are not discussed anywhere in the paper.  Is it common to look at Q/Qexp for different m/z's? I was under the impression that always an overall Q/Qexp is considered in PMF. What is the acceptable range of this

ratio when considering individual m/z's? Values up to 300 are way too high for the overall Q/Qexp. It's concerning that some of the key fragments (43, 55, etc.) have such high values.

- Section 3.1 as written is very confusing. It's mentioned the highest conc. were seen in post-monsoon, but then two periods in post-monsoon are reported. If the point is to discuss the highest concentration, only one number/site/season should be reported. Overall I don't find this beginning paragraph of 3.1 useful as written. If the authors want to describe seasonality, I think the two sites need to be considered separately, so I suggest reorganizing this part.

- L 289: a reference to Chloride deposition is presented here without any discussion of how deposition rate is inferred. Mere reduction in chloride does not mean it's being deposited. Either elaborate or remove this sentence.

- L 363: Aerosol sulfate does not evaporate with typical diurnal changes in temperature whereas ammonium chloride or nitrate do. If oPOA is behaving more like sulfate, I wouldn't call it semi-volatile. Based on the diurnal profiles, I also don't think oPOA has a similar diurnal profile to sulfate! It's more like chloride and nitrate, in which case it's justified to call it semivolatile.

Other concerns:

- Sometime chloride is referred to as Cl and sometimes Cl-. Please use the latter and consistently.

- Sometimes numbers in molecular formulas are not subscripted; please correct.

- This sentence doesn't full make sense: "Comparing these concentrations with the PostM_OD_H2 dataset, HOA concentrations of 30 ug/m3 were identified". Please rephrase.

- Figure S6: data gaps show up with a flat line; please remove those.

- References to figures in SI (PMF section and those after) are not correct; Please correct.

- Fig. S17. Figure caption doesn't match all the panels that are plotted.

---

## Author Comment (AC1) · 30 Mar 2021

**PM$_1$ composition and source apportionment at two sites in Delhi, India across multiple seasons**

Referee 1 comments in black. The response to the comments in orange.

This is what I would consider a "standard" AMS and aethalometer study to document air pollution levels in a major urban area in a part of the world where air quality is among the worst; thus, it does fall within the scope of ACP. Its novelty lies in the application of proven methodology in a new location with a sparse amount of spatial and temporal resolution. It is not ground-breaking, but it is important, as the conclusions are relevant for air quality control efforts in what many consider the developing world. The question then becomes, however – could more be done with the data? On the whole, it is well written and easy to follow (though some care needs to be paid to figure and table numbering), the abstract and title are accurate, and the authors use appropriate citations. The figures are easily read and understood – especially for members of the community who do this sort of work (diurnal profiles, time series, AMSPMF factor spectra).

The authors appreciate the value of the reviewer's comments and the improvements to the quality of the manuscript that have resulted from our responses.

My main concern and why I am unable to recommend publication at this time is the lack of evidence that the instruments are actually reporting comparable data – this is based on a lack of comparison of the AMS spectra and on the lack of agreement (approaching 25% difference) in the one set of collocated data shown. See below.

When comparing the total aerosol concentration measured by the two instruments, the total aerosol concentration for cToF-AMS is 26.5 µg.m$^{-3}$ and for HR-ToF-AMS is 32.1 µg.m$^{-3}$; a 21% difference. This is an acceptable comparison in the context of previous studies that have found uncertainties between 19 – 50 %. The disagreement during the intercomparison is likely a result of a difference in fragmentation or organic relative ionization efficiency but will not affect the factorisation analysis. For instance, Frohlich et al. (2015) in a 15 aerosol mass spectrometer instruments intercomparison, identified f44 to vary by a factor of between 0.6 and 1.3 compared to the mean across all instruments and concluded that no significant influence on the total factor contribution was noticed. The difference of 21% in the present manuscript should be taken to be representative of the uncertainties in the absolute concentrations at the different sites, but this should not affect the diurnal profiles and conclusions of the paper.

The following paragraph has been edited in section S2.1 of the supplement: An intercomparison was performed between the cToF-AMS and the HR-ToF-AMS (fig. S7), deployed at OD over pre-monsoon in order to perform an intercomparison (28/May − 09/June), obtaining average concentrations, in µgm$^{-3}$, of 15.0 and 19.1 of Org, 1.7 and 1.6 of NO$_3^-$, 6.8 and 8.3 of SO$_4^{2-}$, 2.5 and 2.6 of NH$_4^+$, 0.4 and 0.5 of Cl$^-$ for cToF-AMS and HR-ToF-AMS respectively. The total aerosol concentration for cToF-AMS is 26.5 µg.m$^{-3}$ and for HR-ToFAMS is 32.1 µg.m$^{-3}$, a difference of 21%. This is well within the range of previous AMS comparison studies. Crenn et al. (2015) estimated an organic mass uncertainty of 19 %. Bahreini et al. (2009) estimated an overall uncertainty of 35 %, agreeing with other AMS studies (DeCarlo et al., 2008;Dunlea et al., 2009). Recently, a 50% uncertainty has been reported by Shinozuka et al. (2020).

The following paragraph (line 164) has been edited in the main manuscript: In this study, the cToF-AMS and the HR-AMS agree to within 21% from parallel ambient measurements in OD during the pre-monsoon season (Figure S7). Previous comparisons of AMS instruments have shown agreement between instruments of between 19 and 50% and 35% is widely recommended as the absolute accuracy of AMS instruments (DeCarlo et al., 2008;Dunlea et al., 2009;Bahreini et al., 2009;Crenn et al., 2015;Shinozuka et al., 2020).

While there are differences in the response of the two instruments, we do not consider these to undermine the core conclusions of the paper and considered it important to report these as openly as possible, given that most comparable papers only report data from a single instrument without such checks in place. A quantitative difference in the mass concentrations (21%) can be taken as indicative of the accuracy of the comparisons

between the sites. Differences in the precise mass spectral response between instruments has been noted in previous publications, however it has also been shown that this is mitigated through the use of PMF because these are manifested by changes in the factor profiles, not the factor mass concentration time series (Frohlich et al., 2015).

Specific comments Line 154, should Figure 1 be called out here? In fact, Figure 1 is not called out at all until Line 226, though figures should be called out sequentially.

Figure 1 has been referred to in line 150.

Line 165, the C and HR-MS comparison concentrations in S5 do not match the numbers given in the preceding text. In addition, they disagree by~6 ug/m3 (26 for C, 32 for HR), which is a 23% difference.

In figure S5 (now figure S7), the total aerosol concentration for cToF is 26.5 µgm$^{-3}$ and for HR is 32.1 µgm$^{-3}$, a difference of 21%.

To clarify, the comparison shown in figure S5 (now figure S7) shows the data 28/May – 09/June for both cToF and HR instruments. The data presented in the manuscript is 28/May -09/June for cToF and 26/May – 28/June for the HR. The dates of the data used in the manuscript are presented in Table 1. The HR-ToF-AMS was operated by a local researcher, allowing to continue sampling until late June.

**Table 1. Measurement dates and instrumentation. Winter (Win), pre-monsoon (PreM), monsoon (Mon) and post-monsoon (PostM). New Delhi (ND) and Old Delhi (OD) sites. ACSM (A), two HR-ToF-AMS instruments were used (H1 and H2), cToF-AMS (C). Eddy-covariance flux measurements tower (T).**

| Instrument | Event | start | end | Days | Feb | Mar | Apr | May | Jun | Jul | Aug | Sep | Oct | Nov | Dec |
|---|---|---|---|---|---|---|---|---|---|---|---|---|---|---|---|
| ACSM | Win_ND_A | 05/02 | 03/03 | 26 | | | | | | | | | | | |
| HR-AMS_1 | PreM_ND_H1 | 24/04 | 30/05 | 36 | | | | | | | | | | | |
| cToF-AMS | PreM_OD_C | 28/05 | 09/06 | 12 | | | | | | | | | | | |
| HR-AMS_2 | PreM_OD_H2 | 26/05 | 28/06 | 33 | | | | | | | | | | | |
| HR-AMS_2 | Mon_OD_H2 | 03/08 | 19/08 | 16 | | | | | | | | | | | |
| HR-AMS_2 | PostM_OD_H2 | 09/10 | 04/11 | 26 | | | | | | | | | | | |
| cToF-AMS | PostM_OD_C | 11/10 | 15/10 | 4 | | | | | | | | | | | |
| HR-AMS_2 | PostM_OD_T_H2 | 06/11 | 20/11 | 18 | | | | | | | | | | | |
| cToF-AMS | PostM_ND_T_C | 06/11 | 20/11 | 32 | | | | | | | | | | | |

Does this limit the robustness of the spatial comparison made? I actually think this is the most significant weakness of this manuscript. How can the authors justify comparing data from separate locations/times when two of the instruments do not agree when collocated? Were any comparisons done on the PMF results? Are their spectra similar? Do the PMF results actually indicate that the factors being compared have similar characteristics such as O:C, H:C, etc.? Before I can recommend this paper for publication, I need to be convinced that the data are actually comparable, not just told that they are, even if it is in the supplement (line 254). Simply showing the spectra in Figure 4 is insufficient in my opinion.

As mentioned above, a 21% difference is well within the range of previous AMS intercomparisons. The disagreement during the intercomparison is likely a difference in fragmentation or organic relative ionization efficiency but will not affect the factorisation analysis. For instance, Frohlich et al. (2015) in a 15 aerosol mass spectrometer instruments intercomparison, identified f44 to vary 0.6 – 1.3 compared to the mean across all instruments and concluded that the variability on f44 have important influence on the resulting factor profiles but no significant influence on total factor contribution was noticed. The disagreement of 21% in the present manuscript should be taken to be

representative of the errors in the absolute concentrations at the different sites, but this should not affect the diurnal profiles and conclusions of the paper.

A PMF analysis has been performed to the Org concentrations measured with the two AMS instruments, resulting on a considerable good comparison. The following text has been added to section S2.1 of the supplement and figure S7 has been updated as follows:

A PMF analysis was performed to the Org concentrations measured with the cToF-AMS and the HR-ToF-AMS. Figure 7c shows the mass spectra comparison of the factor profiles identified and figure 7.d shows the triangle plot, f44 – f43, to compare and describe OOA. The doted lines represent the space proposed by Ng et al. (2011) to characterise OOA. The parameters f43 and f44 represent the ratio of the integrated signal at m/z 43 and m/z 44 to the total signal in the organic component mass spectrum. The same OA factors were identified in the two PMF analyses, HOA, MO-OOA, BBOA, COA and LO-OOA. This analysis verifies the AMS intercomparison, with the same OA factors and similar ageing (f44-f43).

[Figure]

Figure S7. Average concentrations (S7.a) and relative contribution (S7.b) of Org, NO3-, SO42-, NH4+ and Cl- for the cToF-AMS and the HR-ToF-AMS. Mass spectra comparison from PMF analysis (S7c) and f44-f43 triangle plot (S7.d) to compare OA ageing according to Ng et al. (2011).

Additionally, the triangle plot looking at the f44 – f43 space is plotted for all the datasets. The following text and figure have been added to the supplement. Figure S13 shows the triangle f44 – f43 plot to describe and compare OOA. The dotted lines represent the space proposed by Ng et al. (2011) to characterise OOA. The parameters f43 and f44 represent the ratio of the integrated signal at m/z 43 and m/z 44 to the total signal in the organic component mass spectrum. We can see the typical

behaviour of MOOOA with high f44 values compared to LOOOA, characteristic of a more aged, oxygenated OA. MOOOA, in black, while having different values, is found in a distinct area in the plot with f44 between 0.18 -0.26, while LOOA with low f44 (0.10) and high f43, characteristic of fresher OOA when compared with MOOOA. HOA, in brown, has a low f44, close to zero, and distinct f43 values of 0.08 - 0.13. oPOA, in purple, has slightly high values of both f44 and f43, agreeing with the identification as to be oxygenated primary organic aerosol. This analysis suggests a good separation on the oxygenated species between factor profiles and shows an f43 cluster of HOA. While f44 shows, in general, distinct values for the OA profiles, f43 shows more spread values, considered to be related to the variability on OA sources and processes of the different periods and sites.

[Figure]

Figure S13. f44 vs f43 for all the periods and all the factor profiles and identified with PMF analysis. The symbols represent the sampling periods/sites and the colours define the PMF factor profiles.

Line 179, should this be table S1, not table 1?
Edited text in the manuscript: Table 1 shows details about the instrument locations and sampling periods and table S1 presents the collocated instruments with the mass spectrometers.

Line 201 (and 205 and other places), I would argue that the concentrations observed are lower, not low. 80-200 ug/m3 are still very large concentrations

The changes have been applied as suggested.

Line 245. I believe most AMS data are presented as LO-OOA and MO-OOA (less and more oxidized) rather than as SV and LV, respectively, in more recent literature.

The changes have been applied as suggested.

Section 4.1. I find the discussion on oPOA to be fairly weak, with no justification of the suppositions made. It makes sense that meteorology impacts the oPOA dynamics, then it is mentioned that it tracks sulfate, then it is stated that oPOA may have a similar volatility as NH4Cl. No conclusions are made. I suggest this portion of the discussion be removed or strengthened considerably.

The conclusion about oPOA has been edited and extended as follows: One additional factor (oPOA) had a particular diurnal trend, similar to Cl$^-$, and a mass spectral signature similar to OOA. However, from examination of the polar plots (Fig.S20), oPOA appears to have similar source sectors to $SO_4^{2-}$.

This suggests oPOA may be semi-volatile and driven by changes in T and RH, like Cl-, whilst having different sources, undetermined at this time.

Supplement S2. Please clarify how CE was determined. The paper states that the authors used 0.5 based on the ACSM manual. However, in the plots, some show CE =1, and others show CE = 0.5 (while others show CDCE). In addition, the x-axis in these plots show PM2.5, not PM1. Is that reasonable to use?

The co-authors apologise for not presenting detailed information about the CE determination. Section S2.1 in supplement has been updated with detailed analysis:

S2.1 Calibrations and collection efficiency estimation.

Table S1. Nitrate ion efficiency (IE) and relative IE (RIE) for $NH_4^+$, $SO_4^{2-}$ and Cl- from calibrations performed on the aerosol mass spectrometer instruments. [a]preflux period (11/10/18 - 03/11/18). [b]Diwali period (05/11/18 - 14/11/18). [c]post Diwali (14/11/18 - 23/11/18).

| Instrument | Season | IE | RIE_$NH_4^+$ | RIE_$SO_4^{2-}$ | RIE_Cl- | CE |
|---|---|---|---|---|---|---|
| cToF-AMS | PreM | 1.55E-07 | 4.01 | 1.17 | 1.5 | 0.5 |
| cToF-AMS | PostM | 2.40E-07 | 4.6 | 1.2 | 1.7 | 0.5 |
| HR-AMS_1 | PreM | 3.25E-07 | 4 | 1.31 | 1.3 | 0.5 |
| HR-AMS_2 | PreM | 2.92E-07 | 4 | 1.45 | 2.07 | 0.5 |
| HR-AMS_2 | Mon | 2.92E-07 | 4 | 1.45 | 2.07 | 0.5 |
| HR-AMS_2 | [a]PostM | 2.89E-07 | 4 | 1.45 | 2.07 | 0.5 |
| HR-AMS_2 | [b]PostM | 3.14E-07 | 4 | 1.45 | 1.05 | 0.8 |
| HR-AMS_2 | [c]PostM | 3.14E-07 | 4 | 1.45 | 1.05 | 0.5 |

Collocated $PM_{2.5}$ measurements were performed in a Digitel sampler (DH-77 Digitel Enviro-sense) with a flow rate of 500 L/min. collecting 12-hour samples in quartz fibre filters. The samples were analysed with Ion Chromatography (IC) to measure anion and cation data, including blank subtraction. Components analysed include phosphate, nitrate, bromide, sulphate, nitrite, chloride, fluoride, $K^+$, $Ca^{2+}$, $Mg^{2+}$, $NH4^{2-}$ and $Na^+$. Collected filters were kept frozen and transported to the University of Birmingham for analysis. For the IC analysis, deionized water was used for blank determination and subtraction. 10 mil of DI water were added to samples. Extraction tubes were sonicated for 1 hour with bath temperature not exceeding 27 °C. Next day filter the extract solution for each sample tube using a 10 ml plastic filter and 0.45 μm syringe filter (star labs) into a new labelled polypropylene tube, finally, the sample is ready for IC analysis.

A Partisol (2025i, ThermoFisher Scientific) was deployed to perform 6 hourly gravimetric $PM_{2.5}$ mass. These measurements are used in this manuscript to determine the collection efficiency of the HR-AMS_2.

For HR-AMS_2, a CE = 0.5 was used for preM and postM preflux tower periods, which was determined by comparing AMS+BC with gravimetric $PM_{2.5}$ (Fig. S4.b) and Cl-, $NO_3^-$ and $SO_4^{2-}$ quantified by IC from filter measurements (Fig. S6). In the PostM flux period, for the HR-AMS_2, a CE = 1.0 was derived after comparison with total $PM_{2.5}$ (fig. S4.d). For the HR-AMS_1 measurements, a CE = 0.5 after the intercomparison with the HR-AMS_2 (fig. S7). The ACSM manual recommends using a CE = 0.5.

Figure S5 presents the time series of $PM_1$ online measurements (HR-AMS_2 + BC) and total gravimetric PM2.5 concentrations. Using CE = 0.5 (Fig. 5.b) shows the best agreement between $PM_1$ and $PM_{2.5}$, with a $PM_{2.5}$:$PM_1$ ratio going from 0.8 to 1.4.

[Figure]

Figure S4 Comparison of total PM1 (HR-AMS_2 + BC) with total gravimetric PM2.5 to determine collection efficiency (CE) with HR-AMS_2 and aethalometer (BC) measurements . All AMS+BC measurements are averaged according to filter sampling times.

[Figure]

Figure S5 Analysis of the HR-AMS_2 and total gravimetric PM2.5 for the PostMon preflux period. Time series of averaged PM1 (AMS + BC) (black line) and gravimetric PM2.5 (blue line) for CE =1 (a), CE = 0.5 (b) and CDCE (c). The PM2.5:PM1 ratio is shown in red. All AMS+BC measurements are averaged according to filter sampling times.

[Figure]

Figure S6 Correlations of Cl-, NO3- and SO42- between HR-AMS_2 and filters analysed with ICP-MS by the University of Birmingham for CE = 1, CE = 0.5 and composition dependant CE (CDCE). All AMS+BC measurements are averaged according to filter sampling times.

Supplement S3 and S4. Please check the figure numbering.

The figure numbering has been updated in supplement S3 and S4.

Supplement S5. Please correct the caption for S17, as it shows more than Cl and oPOA.

Citation updated as follows: Figure S20 (before Figure S17). Polar plots of various aerosols. Median concentrations [µg.m$^{-3}$].

Bahreini, R., Ervens, B., Middlebrook, A. M., Warneke, C., de Gouw, J. A., DeCarlo, P. F., Jimenez, J. L., Brock, C. A., Neuman, J. A., Ryerson, T. B., Stark, H., Atlas, E., Brioude, J., Fried, A., Holloway, J. S., Peischl, J., Richter, D., Walega, J., Weibring, P., Wollny, A. G., and Fehsenfeld, F. C.: Organic aerosol formation in urban and industrial plumes near Houston and Dallas, Texas, Journal of Geophysical Research: Atmospheres, 114, https://doi.org/10.1029/2008JD011493, 2009.
Crenn, V., Sciare, J., Croteau, P. L., Verlhac, S., Fröhlich, R., Belis, C. A., Aas, W., Äijälä, M., Alastuey, A., Artiñano, B., Baisnée, D., Bonnaire, N., Bressi, M., Canagaratna, M., Canonaco, F., Carbone, C., Cavalli, F., Coz, E., Cubison, M. J., Esser-Gietl, J. K., Green, D. C., Gros, V., Heikkinen, L., Herrmann, H., Lunder, C., Minguillón, M. C., Močnik, G., O'Dowd, C. D., Ovadnevaite, J., Petit, J. E., Petralia, E., Poulain, L., Priestman, M., Riffault, V., Ripoll, A., Sarda-Estève, R., Slowik, J. G., Setyan, A., Wiedensohler, A., Baltensperger, U., Prévôt, A. S. H., Jayne, J. T., and Favez, O.: ACTRIS ACSM intercomparison – Part 1: Reproducibility of concentration and fragment results from 13 individual

Quadrupole Aerosol Chemical Speciation Monitors (Q-ACSM) and consistency with co-located instruments, Atmos. Meas. Tech., 8, 5063-5087, 10.5194/amt-8-5063-2015, 2015.

DeCarlo, P. F., Dunlea, E. J., Kimmel, J. R., Aiken, A. C., Sueper, D., Crounse, J., Wennberg, P. O., Emmons, L., Shinozuka, Y., Clarke, A., Zhou, J., Tomlinson, J., Collins, D. R., Knapp, D., Weinheimer, A. J., Montzka, D. D., Campos, T., and Jimenez, J. L.: Fast airborne aerosol size and chemistry measurements above Mexico City and Central Mexico during the MILAGRO campaign, Atmos Chem Phys, 8, 4027-4048, 2008.

Dunlea, E. J., DeCarlo, P. F., Aiken, A. C., Kimmel, J. R., Peltier, R. E., Weber, R. J., Tomlinson, J., Collins, D. R., Shinozuka, Y., McNaughton, C. S., Howell, S. G., Clarke, A. D., Emmons, L. K., Apel, E. C., Pfister, G. G., van Donkelaar, A., Martin, R. V., Millet, D. B., Heald, C. L., and Jimenez, J. L.: Evolution of Asian aerosols during transpacific transport in INTEX-B, Atmos. Chem. Phys., 9, 7257-7287, 10.5194/acp-9-7257-2009, 2009.

Frohlich, R., Crenn, V., Setyan, A., Belis, C. A., Canonaco, F., Favez, O., Riffault, V., Slowik, J. G., Aas, W., Aijala, M., Alastuey, A., Artinano, B., Bonnaire, N., Bozzetti, C., Bressi, M., Carbone, C., Coz, E., Croteau, P. L., Cubison, M. J., Esser-Gietl, J. K., Green, D. C., Gros, V., Heikkinen, L., Herrmann, H., Jayne, J. T., Lunder, C. R., Minguillon, M. C., Mocnik, G., O'Dowd, C. D., Ovadnevaite, J., Petralia, E., Poulain, L., Priestman, M., Ripoll, A., Sarda-Esteve, R., Wiedensohler, A., Baltensperger, U., Sciare, J., and Prevot, A. S. H.: ACTRIS ACSM intercomparison - Part 2: Intercomparison of ME-2 organic source apportionment results from 15 individual, co-located aerosol mass spectrometers, Atmos Meas Tech, 8, 2555-2576, 10.5194/amt-8-2555-2015, 2015.

Ng, N., Canagaratna, M., Jimenez, J., Chhabra, P., Seinfeld, J., and Worsnop, D.: Changes in organic aerosol composition with aging inferred from aerosol mass spectra, Atmos Chem Phys, 11, 6465-6474, 2011.

Shinozuka, Y., Saide, P. E., Ferrada, G. A., Burton, S. P., Ferrare, R., Doherty, S. J., Gordon, H., Longo, K., Mallet, M., Feng, Y., Wang, Q., Cheng, Y., Dobracki, A., Freitag, S., Howell, S. G., LeBlanc, S., Flynn, C., Segal-Rosenhaimer, M., Pistone, K., Podolske, J. R., Stith, E. J., Bennett, J. R., Carmichael, G. R., da Silva, A., Govindaraju, R., Leung, R., Zhang, Y., Pfister, L., Ryoo, J. M., Redemann, J., Wood, R., and Zuidema, P.: Modeling the smoky troposphere of the southeast Atlantic: a comparison to ORACLES airborne observations from September of 2016, Atmos. Chem. Phys., 20, 11491-11526, 10.5194/acp-20-11491-2020, 2020.

---

## Author Comment (AC2) · 30 Mar 2021

PM$_1$ composition and source apportionment at two sites in Delhi, India across multiple seasons

Referee 2 comments in black. The response to the comments in orange.

The paper by Reyes-Villegas reports on measurements of aerosol composition and absorption at 2 sites in India (new Delhi and Old Delhi) where ground- based measurements were carried out in different seasons in 2018. Diurnal and seasonal trends in composition and organic aerosol types, resolved by positive matrix factorization, are reported. PM1 composition and concentration differences between the two sites in all seasons are minimal. Organic aerosols (OA) were the dominant component of PM1 in all seasons; higher concentrations of all species were measured in post-monsoon/winter, owing to changes in meteorological conditions. During pre-monsoon, oxygenated OA (LVOOA) was the dominant OA type with peaks appearing during the day, while in post-monsoon and winter periods, hydrocarbon-like OA (HOA) was the dominant aerosol type, with minimum concentrations during the day. Biomass burning OA, oxygenated primary OA, cooking OA, other oxygenated OA (SVOOA) were also identified although not consistently at all sites and all seasons (except for BBOA and COA).

The authors appreciate the value of the reviewer's comments and the improvements to the quality of the manuscript that have resulted from our responses.

Given this is not the first set of measurements in the area (see lines 94-105 for a list of comparable studies) and lack of new scientific insights presented here, I suggest the manuscript be considered as a "Measurement Report".

While AMS observations have previously been made in this region, these have been restricted to single sites and most restricted to a single season. Given the large size and inhomogeneity of sources within developing megacities such as Delhi, it is important that these are not assumed to be spatiotemporally representative. This is the first study to address this question and the systematic multi-dataset PMF analysis and critical comparison of the sites and seasons constitute more than a simply 'measurement' report.

When comparing the two sites, Old Delhi (OD) and New Delhi (ND), no remarkable difference on concentrations was observed; the C-PM$_1$ (C-PM$_1$ = BC + NR-PM$_1$) concentrations were OD (142 +/- 117 μg.m$^{-3}$) compared to ND (123 +/- 71 μg.m$^{-3}$). In some ways it was expected that we would observe a large difference between sites, due to their particular characteristics (OD being a more populated site with potentially more primary sources around and ND being a greener area with parks and business-like buildings in the surroundings). We found though, a large variation on C-PM$_1$ when looking at the seasonal analysis, for example, with C-PM$_1$ concentrations in monsoon to be lower than 60 μg.m$^{-3}$. The following paragraph has been added to conclusions: The fact that there is no remarkable difference on C-PM$_1$ concentrations between the two sites, suggests that, when aiming to control C-PM$_1$ high concentrations, the actions should be regionally orientated, for example as Delhi region, rather than considering controlling air pollution in OD or ND only.

Using PMF and Aethalometer model analysis, Traffic was shown to be the main primary aerosol source for both OA and BC, and as we mention in the conclusions, this indicates that in Delhi further control of primary traffic exhaust emissions would make a significant contribution to reducing PM$_1$ concentrations. Moreover, by controlling gas precursor emissions, OOA concentrations may also be reduced.

These findings are a novel advance, as is recognised by referee 1 and are a valuable addition to the literature. The manuscript provides scientific insight beyond that expected within a "Measurement

Report" and we contend that it should be published on ACP as a research article, after addressing the useful comments the referee has provided.

As such, I still have concerns about the analysis and data quality; some parts of the report also benefit from new reorganization and further clarifications.

Technical concerns:

- What was the sampling protocol/set up at different sites? Was there any cyclone used on the inlet? Was there any control on humidity? If not, how could changes in ambient RH diurnally and between seasons impact the measurements?

A cyclone and a drier were used in all the instruments. Text added to the manuscript: All set ups included a $PM_{2.5}$ cyclone to cut particle size and a drier to reduce humidity.

- Quantification of different AMSs and Figure S4: what is the difference between pre-flux and flux period? Why is the behavior of the comparisons so different between these periods when using the same CE correction factor (0.5 or CDCE)? For pre-flux period, slope with CE=0.5 and CDCE is very similar. It may be that the composition really dictates CE to be close to 0.5, but better justification/explanation needs to be provided as to why 0.5 is used for the whole study and not CDCE. Comparison is done on online PM1 and PM2.5 filter-based data. How/where are these filters collected? No information is provided, yet the measurements serve as the basis for justifying the selected value of the CE! What is the suspected contribution of 1-2.5 um particles to mass that justifies the comparison?

Please see responses to reviewer 1.

- Aethalometer data and contribution of BB vs. traffic: it was not explained why AAE$_{wB}$=2 and AAE$_{tr}$=0.8 is selected. It seems the basis for selecting these factors is the match between the traces in Fig. S13 for the time period in the box, but it wasn't explained why is it expected to have equal absorption from wood burning and traffic at two different wavelengths during Diwali period to justify selecting the default AAEs? I also disagree that there was no change in the absorption calculations using different values of AAE; clearly the b_abs_470wb changes in other panels of this graph depending on the AAE value. Why was a different AAE$_{tr}$ used for AE-33 ND-Winter period (Fig. S15)?

The following paragraph describing the aethalometer analysis has been added to the supplement material (line 199):

A sensitivity test was performed to determine $\alpha_{ff}$ = 0.8. and $\alpha_{bb}$ = 2.0, varying $\alpha_{ff}$ from 0.4 to 2 and $\alpha_{bb}$ from 1.4 to 2.6 and increments of 0.1. Figure S16 shows example plots of the performed analysis. An improvement was observed when using $\alpha_{ff}$ of 0.8 compared to 1.0, thus a value of 0.8 was derived (figures S16.a and S16.b). No significant changes were observed when testing different $\alpha_{bb}$ values (figures S16.c and S16.d), thus the default value of 2.0 was used (Fig. S16). A similar analysis was performed to select values for the subsequent Aethalometer model analyses.

[Figure]

Figure S16. Sensitivity test to select αff = 0.8. The peak marked in panel (a) relates to the Diwali celebrations.

Additional edit. In section S4. Aethalometer analysis, the term $b_{abs\_950tr}$ (traffic) was replaced with $b_{abs\_950\,ff}$ (fossil fuel) and $b_{abs\_470wb}$ (wood burning) with babs_470 bb (biomass burning) to agree with what is presented in the manuscript.

In general, I'm also not sure the way aethalometer data are presented is useful. What would be more useful could be the fraction of absorption at 470 nm that is due to wb vs. traffic and not the absolute value of absorption due to wb at 470 nm.
We are using the Aethalometer model proposed by Sandradewi et al. (2008), which has been widely used to determine the contribution of wood burning and traffic to carbonaceous material using $b_{abs\_470wb}$ and $b_{abs\_950tr}$ respectively. We consider the Aethalometer analysis presented on this manuscript to do a efficient analysis on identifying traffic to be the main carbonaceous source in Delhi, India.

- References to PMF factors for oxygenated OA are given in terms of expected volatility (LV/SVOOA). Isn't it more accurate to call them based on their oxygenation level since the AMS doesn't measure volatility directly?
The terms MO-OOA instead of LVOOA and LO-OOA instead of SVOOA have been used in all the manuscript.

Factors show up in the last panel of Fig. 4 (SFOA and nHOA) that are not discussed anywhere in the paper.
In the last version of the manuscript, in the caption of figure 4, we mention that this dataset is analysed into detail by Cash et al. (2020)

Additional analysis is presented in table S4 where we mention that in order to compare the 7-factor solution selected by Cash et al. (2020) with our 5-factor solution we did the following calculation: **HOA** = HOA_ + NHOA and **BBOA** = SFOA + SVBBOA.

Is it common to look at Q/Qexp for different m/z's? I was under the impression that always an overall Q/Qexp is considered in PMF. What is the acceptable range of this ratio when considering individual m/z's? Values up to 300 are way too high for the overall Q/Qexp. It's concerning that some of the key fragments (43, 55, etc.) have such high values.

The Q/Qexp for different m/z's (for example figure S11) is additional information to look if there is a particular m/z showing a high value. The overall Q/Qexp are also presented in the supplement (for example figure S9). Figure S9 shows Q/Qexp values of around 5.52 for 4-factor solutions and values of around 5.2 for 5-factor solutions. The 6-factor solutions presented two factors with similar time series and mass spectra, characteristic of factor splitting. Hence, the 5-factor solution was chosen to be further analysed. This is explained in S3.

- Section 3.1 as written is very confusing. It's mentioned the highest conc. were seen in post-monsoon, but then two periods in post-monsoon are reported. If the point is to discuss the highest concentration, only one number/site/season should be reported. Overall I don't find this beginning paragraph of 3.1 useful as written. If the authors want to describe seasonality, I think the two sites need to be considered separately, so I suggest reorganizing this part.

This section has been edited in the manuscript comparing only post-monsoon and monsoon as periods with the highest and the lowest concentrations.

Aerosol measurements were taken in OD and ND over different seasons in 2018. The highest concentrations were observed in the post-monsoon season, where OA concentrations of 400 µg.m$^{-3}$ were found (Fig. S8). Lower aerosol concentrations were observed in the monsoon season, where OA reached between 80 – 200 µg.m$^{-3}$.

- L 289: a reference to Chloride deposition is presented here without any discussion of how deposition rate is inferred. Mere reduction in chloride does not mean it's being deposited. Either elaborate or remove this sentence.

We are grateful to the referee for pointing this out, our wording should have referred to concentrations rather than deposition. The line has been edited as follows: This is also consistent with the decrease on Cl$^-$ concentrations before NO3$^-$ concentrations increase in the morning.

Moreover, the following line was added at the end of section 4.1: A more detailed analysis on the Cl$^-$ processes is performed by Gunthe et al. (2021), where the ACSM Winter data is part of the analysed datasets.

- L 363: Aerosol sulfate does not evaporate with typical diurnal changes in temperature whereas ammonium chloride or nitrate do. If oPOA is behaving more like sulfate, I wouldn't call it semi-volatile. Based on the diurnal profiles, I also don't think oPOA has a similar diurnal profile to sulfate! It's more like chloride and nitrate, in which case it's justified to call it semivolatile.

The paragraph has been edited as follows: One additional factor (oPOA) had a particular diurnal trend, similar to Cl$^-$, and a mass spectral signature similar to OOA. However, from examination of the polar plots (Fig.S20), oPOA appears to have similar source sectors to SO$_4^2$. This suggests oPOA may be semivolatile and driven by changes in T and RH, like Cl-, whilst having different sources, undetermined at this time.

Other concerns:

- Sometime chloride is referred to as Cl and sometimes Cl-. Please use the latter and consistently.
The changes have been done as suggested.

- Sometimes numbers in molecular formulas are not subscripted; please correct.
All molecular formulas have been subscripted.

- This sentence doesn't full make sense: "Comparing these concentrations with the PostM_OD_H2 dataset, HOA concentrations of 30 ug/m3 were identified". Please rephrase.

The paragraph has been rephrased as follows: HOA is the factor with the highest concentration with $6.0 – 55.0$ $\mu g.m^{-3}$ for Pre-monsoon and Post-monsoon, respectively, in the PostM_OD_H2 dataset (Table S4), HOA concentrations of 30 $\mu g.m^{-3}$ were identified. These HOA concentrations represent a relative contribution to OA sources of 20 to 50%.

- Figure S6: data gaps show up with a flat line; please remove those.

Figure S6 (now figure S8) has been updated.

- References to figures in SI (PMF section and those after) are not correct; Please correct.
The references to figures have been corrected.

- Fig. S17. Figure caption doesn't match all the panels that are plotted.

Figure 17 (now figure S20) has been updated as follows: Figure S20. Polar plots of various aerosols. Median concentrations [$\mu g.m^{-3}$].

Cash, J. M., Langford, B., Di Marco, C., Mullinger, N., Allan, J., Reyes-Villegas, E., Joshi, R., Heal, M. R., Acton, W. J. F., Hewitt, N., Misztal, P., Drysdale, W., Mandal, T. K., Shivani, Gadi, R., and Nemitz, E.: Seasonal analysis of submicron aerosol in Old Delhi using high resolution aerosol mass spectrometry: Chemical characterisation, source apportionment and new marker identification, Atmos. Chem. Phys. Discuss., 2020, 1-42, 10.5194/acp-2020-1009, 2020.
Gunthe, S. S., Liu, P., Panda, U., Raj, S. S., Sharma, A., Darbyshire, E., Reyes-Villegas, E., Allan, J., Chen, Y., Wang, X., Song, S., Pöhlker, M. L., Shi, L., Wang, Y., Kommula, S. M., Liu, T., Ravikrishna, R., McFiggans, G., Mickley, L. J., Martin, S. T., Pöschl, U., Andreae, M. O., and Coe, H.: Enhanced aerosol particle growth sustained by high continental chlorine emission in India, Nat Geosci, 14, 77-84, 10.1038/s41561-020-00677-x, 2021.
Sandradewi, J., Prévôt, A. S. H., Szidat, S., Perron, N., Alfarra, M. R., Lanz, V. A., Weingartner, E., and Baltensperger, U. R. S.: Using aerosol light abosrption measurements for the quantitative determination of wood burning and traffic emission contribution to particulate matter, Environmental Science and Technology, 42, 3316-3323, 10.1021/es702253m, 2008.

---

## Author Response (AR2)

Reviewer 2 comments response in orange

I appreciate the efforts from the authors to address my previous comments. Given the recommendation of the other reviewer and the authors highlighting the value of continuous, multi-site study in New and Old Delhi, I am fine with considering the paper as a research article. There are however still deficiencies that need to be addressed before acceptance:

The authors appreciate the time spent by the reviewer to provide the valuable comments to our manuscript.

- Indicate the inlet humidity after use of the drier (and was it a nafion drier or a diffusion drier, etc.)

The drier used was a nafion drier. The humidity was not continuously monitored, because the purpose was just to stop the humidity going >80% when going to the instruments.

- My question related to the high Q/Qexp for some m/z's isn't answered. So what does it mean that some of the key fragments have very large Q/Qexp even if the overall Q/Qexp is not too high? Is a value of 300 not high enough to be of concern?! If the overall Q/Qexp is acceptable, and m/z-dependent Q/Qexp is not going to be discussed, please remove the plots.

Thank you for your comment, following your suggestion: Figure S10. Time series of residuals and Q/Qexp values. And Figure 11 Residuals and Q/Qexp values for m/z have been removed.

- Figure S4. Change "Birmingham PM2.5" to "Gravimetric PM2.5" to be consistent with the text

The figure has been updated as suggested.

- By "6 hourly gravimetric PM2.5 mass" do you mean 6-hr filters? As written it seems 6 of 1-hr filters, but the plots have more than 6 filter values, so this needs to be clarified.

We mean 6-hour filters. This has been corrected in the supplement

- Figure S6 indicates anions were measured by ICP-MS. I don't think that's correct. Do the authors mean IC?! Again, remove the reference to Birmingham.

Thank you for the comment. Yes, ions were measure by Ion chromatography, this has been corrected in the supplement.

- Figure 4. Factor nHOA is used in Fig. 4 while NHOA is used in Table S4. Use them consistently please. Also, Figure 4 caption need to have a similar description to what's included for Table S4.

The changes have been done as suggested.

---

## Author Response (AR3)

Editor comments response in orange

Thank you for your response to the comments by Reviewer 2. I think all the comments have been addressed but some additional info and explanation regarding the Q/Qexp question would be useful. Specifically, as the reviewer noted, some individual m/z have very high Q/Qexp, up to 300, but the overall Q/Qexp is only around 5. It is unclear why this is the case. It appears that the individual m/z having high Q/Qexp are HOA type ions. Have you checked the input errors for PMF manually, and evaluated if these input errors are too low? If not, it would be useful to double check the errors of these m/z manually.

All these info would be useful for the readers to know to better understand how the PMF solution was obtained. Please explain these observations (very high Q/Qexp for some m/z, but an overall Q/Qexp of around 5), you can keep Figures S10 and S11 in the SI, and add the explanations to the figure captions.

We appreciate the effort from the editor to provide these comments that will increase the quality of the manuscript.

We looked into the input errors for m/z 55 and m/z 57 and they do not have too low values.

We use the Source Finder (SoFi version 4.8) tool (Canonaco et al., 2013) to run PMF. SoFi gives three types of Q/Qexp values to look into detail on the solutions. One overall Q/Qexp, it also gives a row (time series) Q/Qexp and a column (m/z) Q/Qexp. The overall Q/Qexp is calculated as the mean of the time series Q/Qexp.

We looked into detail on the Q/Qexp values for all the PMF analyses and also PMF analyses from previous projects. The overall Q/Qexp and the time series Q/Qexp are correct. However, in all the analysed datasets the m/z Q/Qexp values are similarly high. We identified that dividing the values in figure S11 by the Q/Qexp and by the number of m/z variables we estimate a value similar to the overall Q/Qexp. We consider SoFi 4.8 might have a bug or missed a calculation to estimate the Q/Qexp for m/z. Currently there is a new version of SoFi 6.0, we will see if the Q/Qexp for m/z are reported correctly, otherwise we will contact the developers. We are using the old SoFi 4.8 because we have developed our own code to explore the solution space from runs perfomed in SoFi 4.8.

We took the idea of dividing by Qexp divided by number of species from the EPA PMF 5.0 fundamentals user guide that states: "the Q/Qexp for a species is the sum of the squares of the scaled residuals for that species, divided by the overall Qexpected divided by the number of strong species". The change we are doing to figure S11 is change the label Q/Qexp for Squares of scaled residual.

We calculate the sum of the values (Squares of scaled residual) plotted in Figure S11 for the specific PMF run. For example, for the optimal solution (PMF_5F_S2) the sum = 11269.56.

We calculate the Qexp = $n * m - p*(n + m)$ = 147127 for the 5-factor solution, where

n = num of samples = 2169, m = num of m/z = 73, p = num of factors = 5

We also divide the Qexp by m, which gives 2015.44.

Finally, Q/Qep = 11269.56/2015.44 = 5.59, which is close to the overall Q/Qexp = 5.2.

We did a similar estimation for other datasets; for example, for the HR-AMS in ND preMonsoon we estimated a new Q/Qexp = 1.70, with n = 40542, m = 77 and p = 5, when the overall Q/Qexp = 1.60.

We have updated the scale of figure S11 for from 300 to 600 and we can see that the highest values are for m/z 15 and m/z 38 (around 550), while m/z55 and m/z57 have values of around 350. Figures S10 and S11 are used to identify particularly high Q/Qexp values for m/z or episodes on time series that might require attention. We are more interested on the relative contribution rather than on the absolute values, hence we consider that this change on figure S11 does not impact on the selection of the optimal solutions. For example, we can see an improvement on both time series and m/z plots when going from 4-factor to 5-factor solutions. However, no remarkable improvement between 5-factor solution was observed. Hence, we can use the overall Q/Qexp to select the optimal solution.

**We have kept figures S10 and S11 in the SI as suggested with the following explanation:**

We added this paragraph to the SI in line 133

The PMF analysis was performed using the the SoFi tool version 4.8 (Canonaco et al., 2013). In order to select the optimal PMF solution, we analysed the overall Q/Qexp (Figure S9), the row (time series) Q/Qexp (Figure S10) and the Squares of scaled residual (Figure S11).

We added this paragraph to the SI in line 150

The overall Q/Qexp =5.2 is calculated as the mean of the time series Q/Qexp. The overall Q/Qexp can also be estimated from the Squares of scaled residuals. We calculate the sum of the values (Squares of scaled residual) plotted in Figure S11 for the specific PMF run. For example, for the optimal solution (PMF_5F_S2) the sum = 11269.56.

We calculate the $Q_{exp}$ = n * m - p*( n + m) = 147127 for the 5-factor solution, where

n = num of samples = 2169, m = num of m/z = 73, p = num of factors = 5

We also divide the $Q_{exp}$ by m, which gives 2015.44.

Finally, Q/Qep = 11269.56/2015.44 = 5.59, which is close to the overall Q/Qexp = 5.2.

Figures S10 and S11 are used to identify particularly high Q/Qexp values for m/z or episodes with high Q/Qexp on time series that might require further analysis. For example, we can see an episode with high Q/Qexp values on 20/20/2018 (Figure S10), which is related to an episode with high Org concentrations. Also, we can see an improvement on both time series (Figure S10) and m/z (Figure S11) plots when going from 4-factor to 5-factor solutions. However, no remarkable improvement between 5-factor solution was observed. Hence, we can use the overall Q/Qexp to select the optimal solution.

However, if the editor considers these details could confuse the reader, we can remove Figure S11 from the supplement.

[Figure]

Figure S1. Time series of residuals and Q/Qexp values.

[Figure]

Figure S2. Residuals and Squares of scaled residual for m/z.

Canonaco, F., Crippa, M., Slowik, J. G., Baltensperger, U., and Prevot, A. S. H.: SoFi, an IGOR-based interface for the efficient use of the generalized multilinear engine (ME-2) for the source apportionment: ME-2 application to aerosol mass spectrometer data, Atmos Meas Tech, 6, 3649-3661, DOI 10.5194/amt-6-3649-2013, 2013.